# PhysReaction: Physically Plausible Real-Time Humanoid Reaction Synthesis via Forward Dynamics Guided 4D Imitation

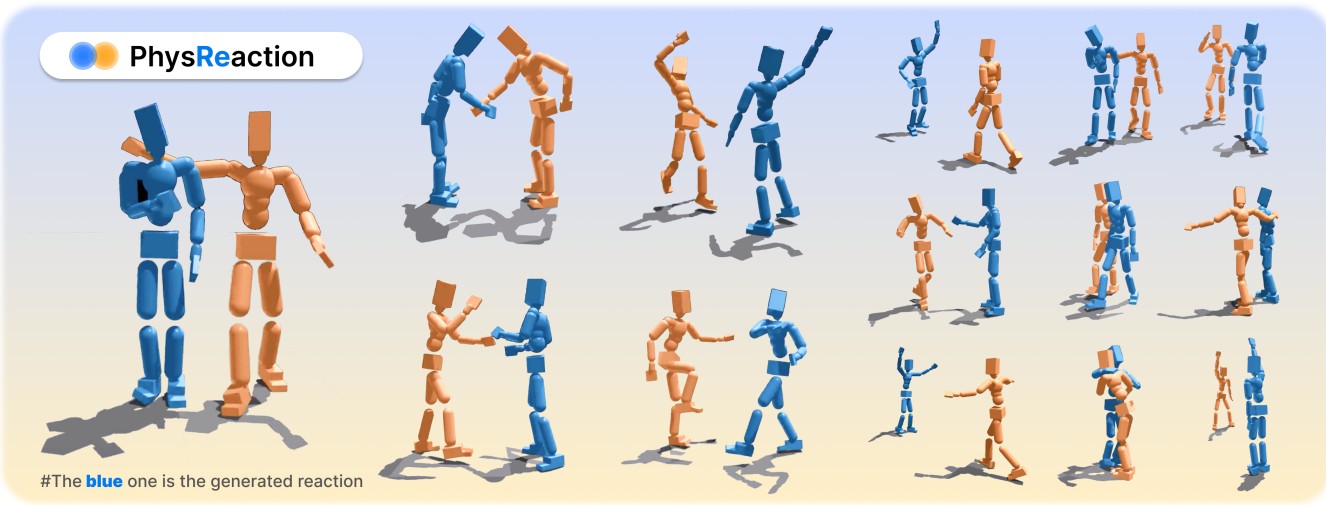

#The **blue** one is the generated reaction

**Figure 1: We introduce the Forward Dynamics Guided 4D Imitation method, a novel approach that employs a neural model to simulate human forward dynamics in real-time at 30 fps(speed up x33). This model guides the process of 4D imitation learning, enabling the generation of reactions that are not only physically plausible but also closely mimic human behavior.**

## ABSTRACT

Humanoid Reaction Synthesis is pivotal for creating highly interactive and empathetic robots that can seamlessly integrate into human environments, enhancing the way we live, work, and communicate. However, it is difficult to learn the diverse interaction patterns of multiple humans and generate physically plausible reactions. Currently, the predominant approaches involve kinematics-based and physics-based methods. The kinematic-based methods lack physical prior limiting their capacity to generate convincingly realistic motions. The physics-based method often relies on kinematics-based methods to generate reference states, which struggle with the challenges posed by kinematic noise during action execution. Moreover, these methods are unable to achieve real-time inference constrained by their reliance on diffusion models. In this work, we propose a Forward Dynamics Guided 4D Imitation method to generate physically plausible human-like reactions. The learned policy is capable of generating physically plausible and human-like reactions in real-time, significantly improving the speed(x33) for inference

and quality of reactions compared with the existing methods. Our experiments on the InterHuman and Chi3D datasets, along with ablation studies, demonstrate the effectiveness of our approach. More visualizations are available in supplementary materials.

## CCS CONCEPTS

• **Computing methodologies** → *Activity recognition and understanding*; *Motion processing*; *Physical simulation*.

## KEYWORDS

Physics-based Animation, Reaction Synthesis, Imitation Learning

## 1 INTRODUCTION

Humanoid Reaction Synthesis refers to the technology involved in enabling humanoid robots to mimic human reactions in a way that feels authentic and natural. The ultimate goal is to achieve a level of interaction where robots can engage with humans seamlessly, understanding and responding to social cues, emotions, and environmental factors just as a human would. The development of humanoid reaction synthesis represents notable progress in robotics and artificial intelligence, aiming toward the creation of robots that could enhance their roles in social contexts, such as acting as assistants, support systems, or team members in various settings. The enhancement of robots' capabilities to interpret and emulate human reactions plays a crucial role in generating smoother and more practical interaction between humans and robots.

Previous research[3, 10, 11, 18, 22, 40, 57, 58, 60–64] has largely concentrated on generating images of individual humans and their interactions with objects. More recently, there has been a shift towards investigating the text-conditioned generation of interactions [25, 57] involving two humans, exemplified by studies like InterGen[25]. Another line of method[5, 23, 44] for generating human motions leverages physics, utilizing training within simulation environments to ensure that the produced motions comply with actual physical laws. A standout example in this domain is InsActor[44], a text-driven, physics-based approach to human motion generation. However, existing methods have the following limitations. **Firstly**, the kinematics-based approaches face challenges, including issues like floating feet, sliding, penetration, and other problems that defy physical plausibility. The absence of physical priors in these methods often hampers their ability to produce convincingly realistic motions. **Secondly**, the existing physics-based method often relies on kinematics-based methods to generate reference states, which struggle with the challenges posed by kinematic noise during action execution. For example, InsActor starts by using a kinematics-based diffusion model to generate a reference state. Actions are then derived from these reference states and the current state of motion. If the kinematics-based diffusion model in InsActor produces the reference state with noise, generating reasonable reactions based on these noisy reference states and the current state becomes unfeasible. **Thirdly**, both of the above methods cannot be directly extended to the reaction synthesis task, because in reality, reaction synthesis is an online setting, meaning actions beyond the current moment are not known. Moreover, practical scenarios demand the capacity for real-time prediction, yet both discussed methods rely on diffusion models, which inherently lack the capability for real-time inference.

Our key idea is to learn the mapping between interaction states directly and reactor actions bypassing the need for a reference state. Consequently, this approach can generate physically plausible reactions while avoiding the noise impact from kinematics-based methods. Moreover, instead of using kinematics-based diffusion models, our method utilizes lightweight networks to achieve real-time inference at 30 fps(speed up x33), making its deployment on real robots possible. Besides, we model the problem as an online setting, which was not considered in previous methods.

Specifically, we introduce a Forward Dynamics Guided 4D Imitation method aimed at developing a reactor policy for synthesizing reactions. This method takes as input the current actions of both the actor and the reactor, along with the next state of the actor, to determine the action the reactor should execute at the current moment. However, directly learning this mapping is not a trivial task, as minor variations in actions can result in drastically different outcomes. To address this, we propose employing a Forward Dynamics Model to guide the imitation learning process, thereby establishing a stable correlation between states and actions. Our approach is structured into four main components: Demonstration Generation Process, Forward Dynamics Model Training, Iterative Generalist-Specialist Learning Strategy, and Forward Dynamics Guided 4D Imitation Learning. We employ a universal motion tracker to convert motion capture data in the simulation environment seamlessly for demonstration generation. Subsequently, our Forward Dynamics

Guided 4D Imitation method, coupled with an Iterative Generalist-Specialist Learning Strategy, is deployed to train the final reactor policy for reaction synthesis.

To evaluate the efficacy of our approach, we conducted experiments on the InterHuman[25] and Chi3D[13] datasets. Our method consistently and significantly surpasses the previous methods across all evaluated metrics. Unlike kinematic-based methods, our approach can produce physically plausible reactions. Moreover, when compared to the InsActor method, our method effectively mitigates the influence of kinematic noise on policy learning, facilitating the establishment of a stable relationship between states and actions. We also conducted comprehensive ablation experiments to verify the effectiveness of each component. Analytical experiments further demonstrate our method's superior performance, especially in terms of resistance to noise and efficiency with small training datasets.

The key contributions of this paper are threefold: i) We introduce a new task focused on the physics-based online synthesis of humanoid reactive motions; ii) We present a novel approach, the Forward Dynamics Guided 4D Imitation, designed to produce realistic and physically plausible reactions, enabling real-time at 30 fps(speed up x33) and online inference; iii) Experiments on InterHuman and Chi3D demonstrate that our method significantly outperforms existing methods. Additionally, detailed ablation studies and analytical experiments have been conducted to prove the effectiveness of our approach.

## 2 RELATED WORK

### 2.1 Human Reaction Synthesis

Some recent research [25, 45, 47, 57] have shifted focus towards the synthesis of human-human interactions. [25] introduced a dataset featuring natural language descriptions and developed a diffusion model to generate human-human interactions. However, their approach faces limitations in reaction synthesis due to its reliance on a fixed CLIP branch for text feature extraction. In contrast, [57] introduced a GAN-based Transformer designed for action-conditioned motion generation. Our research pivots towards generating motions that are conditioned on the movements of another human. This focus is particularly crucial for generating human reactions based on an actor's motion, a key aspect in advancing VR/AR technologies and humanoid robotics, where generating appropriate responses is essential. [9] have proposed a Transformer network that incorporates both temporal and spatial attention mechanisms to generate reactions, while [4] have focused on predicting human intent in human-to-human interactions. Nevertheless, these methods primarily address the generation of body motions without considering the physical properties of those movements. [44] introduce InsActor, a text-driven, physics-based methodology for human motion generation that initiates with a kinematics-based diffusion model to create reference trajectories. Subsequent actions are obtained from the reference state and the current state. While this method depends on an accurate reference state, it struggles with the challenges posed by kinematic noise during action execution, which can adversely affect policy learning. Our approach distinguishes itself as the first to propose a physics-based method for reaction

generation, effectively immune to the noise issues associated with kinematics-based methods.

## 2.2 Imitation Learning and Policy Distillation

Previous studies have leveraged imitation learning techniques, such as behavior cloning [24, 49], enriching Reinforcement Learning with augmented demonstrations [12, 41, 42, 46, 56], and employing Inverse Reinforcement Learning [1, 14, 20, 26, 33] to capitalize on expert demonstrations or policies. Several approaches [16, 21, 32, 48] have embraced the *Generalist-Specialist Learning* concept, wherein a cohort of specialists (teachers) is trained on distinct segments of the task spectrum. Subsequently, their knowledge is distilled into a single generalist (student) across the entire task domain, employing the aforementioned imitation learning and policy distillation techniques. In this work, we introduce a Forward Dynamics Guided Imitation approach, incorporating an iterative Generalist-Specialist learning strategy to train the reactor policy.

## 2.3 Motion Tracking

Simulated characters, constrained by physics [6, 8, 15, 17, 19, 31, 34, 36–39, 52, 53, 59], excel in generating natural human motions and interactions, both human-to-human [27, 55] and human-object [31, 38]. However, the non-differentiability of most physics simulators necessitates the use of time-intensive and expensive reinforcement learning (RL) for training. Motion imitators have shown remarkable capability in mimicking reference motions, especially with high-quality MoCap data, but primarily in smaller datasets. Innovations like ScaDiver [54] and MoCapAct [50] have made significant strides in scaling imitation to larger datasets, achieving up to 80% effectiveness. UHC [29] notably imitates 97% of the AMASS dataset, with its successor, PHC [28], improving on this by eliminating the need for external forces. Our work leverages PHC [28] for high-fidelity motion capture in simulations, transforming the motion capture data into state-action pairs for policy learning. The training result on these state-action pairs showcases the superior performance of our physics-based method over kinematics-based motion tracking in reaction imitation.

## 3 PROBLEM FORMULATION

In this work, we aim to develop a universal reactor policy that enables reasonable social interactions, derived from the observation of an actor's state and its state. We achieve this by learning from a broad spectrum of multi-human interaction scenarios. The actor-reactor interaction, denoted as $\mathbf{x}$, is represented as a collection of motion trajectories $\mathbf{x}_{h \in (act, react)}$, such that $\mathbf{x} = \{\mathbf{x}_{act}, \mathbf{x}_{react}\}$, with $\mathbf{x}_{h \in (act, react)} = \{s_i^h, a_i^h\}_{i=1}^L$ comprising a sequence of state-action pairs.

**State.** The simulation state, denoted by $s_t^{sim} \triangleq (s_t^{act}, s_t^{react}, s_{t+1}^{act})$, captures the actor's state at times $t$ and $t+1$, along with the reactor's state at time $t$. Human states are defined through joint positions $p_t \in \mathbb{R}^{J \times 3}$ and velocities $\dot{p}_t \in \mathbb{R}^{J \times 3}$, with all coordinates recalibrated to the reactor's reference frame based on their current orientation and root position.

**Action.** We employ a proportional derivative (PD) controller for each degree of freedom (DoF) of the reactor, with the action $a_t^{react}$ setting the PD target. The torque applied at each joint is calculated as $\tau^i = k^p \circ (a_t^{react} - s_t^{react}) - k^d \circ \dot{s}_t^{react}$. Building upon the PHC framework, the SMPL body model comprises 24 rigid bodies, 23 of which are actuated, thus defining an action space $a_t^{react} \in \mathbb{R}^{23 \times 3}$.

To set up the environment, we initialize the actor and reactor at the starting state of their trajectory. Our objective, using the simulation state $s_t^{sim} \triangleq (s_t^{act}, s_t^{react}, s_{t+1}^{act})$ at time $t$, is to produce an appropriate action $a_t^{react}$ that facilitates reaching the subsequent actor state $s_{t+1}^{react}$. This scenario presents a multi-task policy learning challenge without specific reward mechanisms, necessitating that our learned policy demonstrates robust generalization across various multi-human interaction tasks.

## 4 METHOD

This section provides a detailed description of our proposed methodology. section 4.1 introduces the framework and training pipeline. The process for generating demonstrations from motion capture data is described in section 4.2, along with the training procedure for the forward dynamics model in section 4.3. Additionally, we leverage a Forward Dynamics Model in section 4.4 to guide 4D Imitation Learning and adopt an Iterative Generalist-Specialist Learning strategy in section 4.5.

## 4.1 Overview

Our methodology encompasses four key components: Demonstration Generation Process, Forward Dynamics Model Training, Iterative Generalist-Specialist Learning Strategy, and Forward Dynamics Guided 4D Imitation Learning, as depicted in fig. 2.

To model state-action relationships, it's crucial to associate actions $a_t$ with each state $s_t$. Thus, during the demonstration generation phase, we employ a universal motion tracker [28] to seamlessly convert motion capture data for use in the simulation environment. This imported data undergoes meticulous manual review to guarantee the demonstration's quality.

A proficient policy should anticipate the outcomes of its actions, to be specific, the future states. Traditional imitation learning struggles with dynamic perception, mainly capturing the current state without forecasting future states. We propose the Forward Dynamic Model to predict the future states. After obtaining state-action pairs $\{s_i^h, a_i^h\}_{i=1}^L$, we initially train two Variational Autoencoders (VAE) as feature extractors for both states and actions, termed the state VAE and the action VAE. Following this, a forward dynamics model is trained to estimate the upcoming state $s_{t+1}$, based on the current state $s_t$ and action $a_t$. We train the model in the feature space using the contrastive loss, as a result, the forward dynamics model is stochastic rather than deterministic leading to more diverse and accurate simulation.

With the forward dynamics model, we advance to the training phase of 4D imitation learning. The term "4D imitation learning" reflects the incorporation of temporal data in our input states and the dynamics network's ability to forecast future states. This approach transcends basic one-to-one mapping, evolving from singular state-action relationships to encompass temporal progression. Utilizing the previous states of both the actor and reactor $\{s_t^{act}, s_t^{react}\}$ along with the actor's current state $s_{t+1}^{act}$, our model is tasked with forecasting the action $a_t^{react}$ for the reactor to execute. This model anticipates the reactor's next state $s_{t+1}^{react}$, leveraging the reactor's

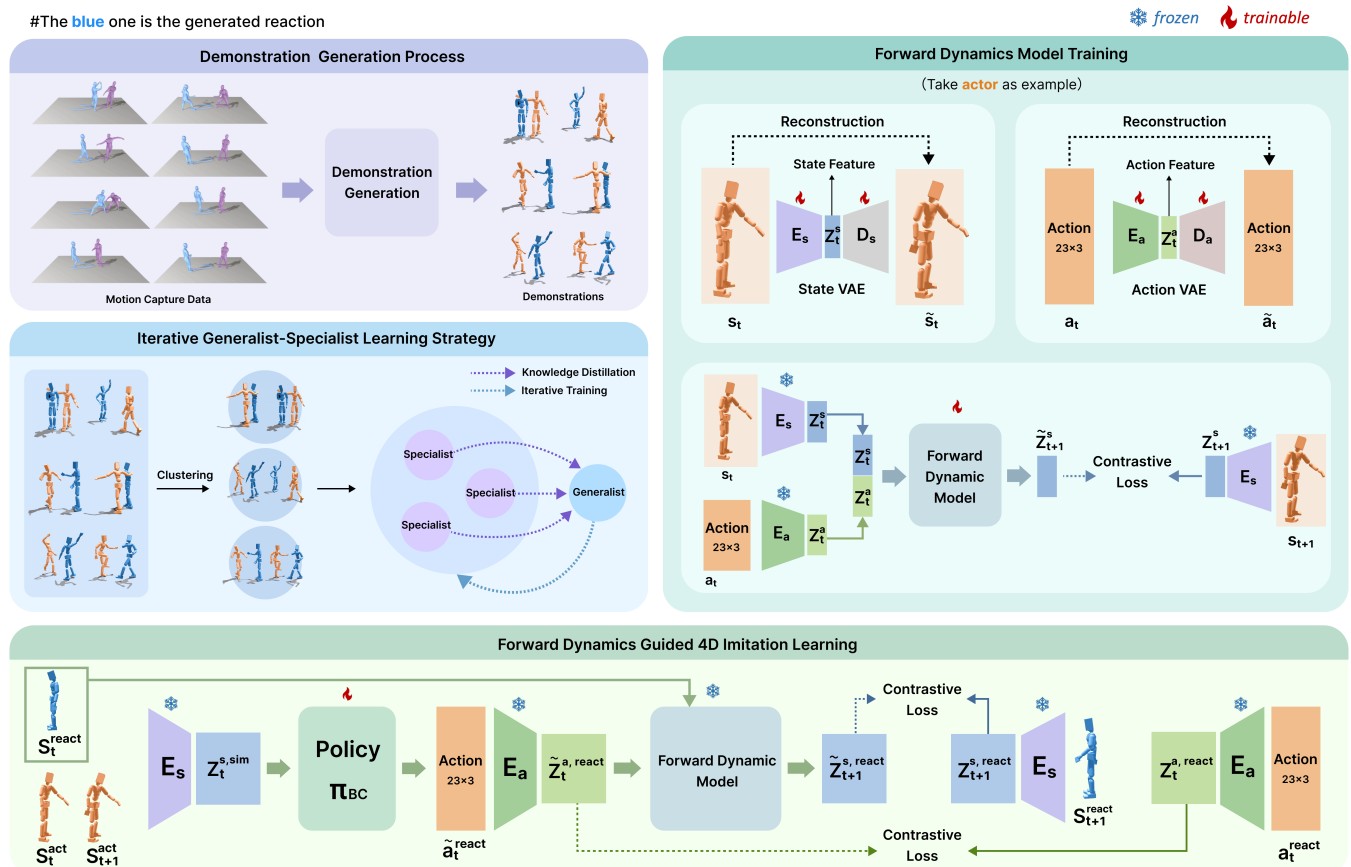

**Figure 2: Our method can be divided into four parts: Demonstration Generation Process, Forward Dynamics Model Training, Iterative Generalist-Specialist Learning Strategy, and Forward Dynamics Guided 4D Imitation Learning.**

preceding state $s_t^{react}$ and the proposed action $a_t^{react}$. After encoding the forecasted action and state through the VAE's encoder, we apply a contrastive loss to facilitate gradient back-propagation.

Given the complexity of imitating a diverse array of tasks with a single network, inspired by [51], we utilize an Iterative Generalist-Specialist Learning Strategy during the imitation learning phase. We begin by clustering dataset motions into ten subsets based on state features from the state encoder. A Generalist model is first trained on the entire dataset, after which this model is duplicated ten times to specialize in each subset, creating ten Specialists. Subsequently, we apply a data distillation technique to transfer the knowledge from these Specialists back to the Generalist. This iterative process enhances our policy's ability to handle a broad spectrum of interactive tasks, enabling the generation of different reactions.

## 4.2 Demonstration Generation from Motion Capture Datasets

To enhance imitation learning, transforming motion capture data into state-action pairs is crucial. However, deriving precise actions from motion data is typically difficult, necessitating high-precision force sensors or advanced motion-tracking techniques. Our goal is to generate accurate state-action pairs $\{s_i, a_i\}_{i=1}^{L}$ from sequences of joint positions $\{p_t\}_{i=1}^{L}$.

In contrast to approaches like DeepMimic[35] or DiffMimic[43], which train a distinct policy for each motion sequence, PHC[28] develops a unified policy adept at tracking various motion sequences. This methodology offers significant efficiency, allowing for direct prediction and greatly enhancing the process of transforming motion capture data into simulations.

Within the PHC framework, a goal-conditioned policy $\pi_{PHC}$ aims to mimic reference motion capture data $\{p_t\}_{i=1}^{L}$, modeling the task as a Markov Decision Process (MDP) defined by the tuple $\mathcal{M} = \langle S, A, T, \mathcal{R}, \gamma \rangle$, encompassing states, actions, transition dynamics, reward function, and discount factor. The objective is to optimize the cumulative discounted reward $\mathbb{E}\left[\sum_{t=1}^{T} \gamma^{t-1} r_t\right]$ through proximal policy gradient (PPO) learning. The control policy $\pi_{PHC}(a_t|s_t) = \mathcal{N}(\mu(s_t), \sigma)$ is described by a Gaussian distribution with a fixed diagonal covariance. This tracking policy is trained on AMASS, a comprehensive motion dataset.

Utilizing PHC as a motion tracker facilitates the import of motion capture data into the simulation environment, yet the tracker's inherent randomness and capability constraints may lead to inaccuracies in tracking complex or intense movements. To counteract this, we track the same dataset 10 times, selectively curating

high-quality results for demonstrations. This approach achieves an overall tracking success rate of approximately 50%, underscoring the tracker's role in not only importing but also physically and reliably augmenting the dataset, thereby enlarging the training set's scale. The imported dataset undergoes meticulous manual review to guarantee the demonstration's quality.

## 4.3 Forward Dynamics Model Training

A proficient policy should anticipate the outcomes of its actions, necessitating a forward dynamics model to enable complex control tasks through learning from experience. Training agents to learn dynamics from intricate, high-dimensional data such as pose observations brings a significant challenge. Rather than predicting directly in the pose space, we opt to translate pose observations into a feature space. The forward dynamics model is trained to maximize the similarity between the predicted and the observed next-state representation. Consequently, our preliminary phase entails the training of both a state encoder and an action encoder to convert raw signals into a comprehensible feature space.

**Representation model for state and action.** We employ a state Variational Autoencoder (VAE), comprising a state encoder $\mathcal{E}_s$ and a state decoder $\mathcal{D}_s$, to transform a state $s_t$ into a state feature $z_t^s$, and then reconstruct $\tilde{s_t} = \mathcal{D}(z_t^s) = \mathcal{D}(\mathcal{E}(s_t))$ from $z_t^s$. Similarly, an action VAE, with an action encoder $\mathcal{E}_a$ and action decoder $\mathcal{D}_a$, is trained to produce the action feature $z_t^a$.

**Forward Dynamics Model Training.** The forward dynamics model forecasts the feature of the subsequent state $z_{t+1}^s$ using the current state $z_t^s$ and action $z_t^a$. Representing state-action-state sequences as $(z_t^s, z_t^a, z_{t+1}^s)$, the model function is $\tilde{z}_{t+1}^s = F(z_t^s, z_t^a)$, encapsulating our forward dynamics model as:

$$\tilde{z}_{t+1}^s = F(z_t^s, z_t^a), \tag{1}$$

where $z_t^s = \mathcal{E}(s_t), z_t^a = \mathcal{E}(a_t)$.

**Contrastive Loss as alignment score.** Optimizing the forward dynamics model to strictly match $z_{t+1}^s$ with $\tilde{z}_{t+1}^s$ presumes deterministic transitions, an assumption not always valid in the dynamic real-world scenarios. Rather than requiring exact matches for zero cost in the cost function, the energy-based contrastive loss permits low costs for all compatible prediction-observation pairs. We select a mini-batch of $N$ state-action-state tuples $(z_t^s, z_t^a, z_{t+1}^s)$. Within this framework, a prediction $\tilde{z}_{t+1}^s$ and its ground-truth $z_{t+1}^s$ from the same tuple are considered a positive pair, whereas other tuple combinations within the mini-batch serve as negative examples. Cosine similarity measures the distance between two representations, with the loss for a positive example pair $(\tilde{z}_{t+1}^i, z_{t+1}^i)$ is defined accordingly:

$$L_f = -\log \frac{\exp(\text{sim}(\tilde{z}_{t+1}^{s,i}, z_{t+1}^{s,i})/\tau)}{\sum_{\substack{j=1 \\ j \neq i}}^{N} \exp(\text{sim}(\tilde{z}_{t+1}^{s,i}, z_t^{s,j})/\tau) + \sum_{\substack{j=1 \\ j \neq i}}^{N} \exp(\text{sim}(\tilde{z}_{t+1}^{s,i}, z_{t+1}^{s,j})/\tau)}, \tag{2}$$

where $\tau$ denotes a temperature parameter that is 0.07 as same as MoCov2[7].

## 4.4 Forward Dynamics Guided 4D Imitation

Our goal is to learn the mapping function from state space to action space for effective and human-like interactions. Reinforcement Learning (RL) techniques, while powerful, often require exhaustive training and can suffer from instability across various scenarios. Traditional imitation learning struggles with dynamic perception, mainly capturing the current state without forecasting future states, leading to unrealistic movements and high-frequency jitter. We introduce a novel forward dynamics-guided 4D imitation learning strategy to address these challenges.

Consider a stochastic MLP policy $\pi_{\text{BC}}(a_t^{react}|s_t^{sim})$ with parameters $\theta_{\text{BC}}$, where $s_t^{sim} \triangleq (s_t^{act}, s_t^{react}, s_{t+1}^{act})$ defines the simulation state. This policy determines the reactor's action $a_t^{react}$. Subsequently, we apply the trained action encoder to convert actions into feature space representations $z_t^{a,react}$. Echoing the forward dynamics model's training, contrastive loss is employed for policy learning supervision, simplifying $z_t^{a,react}$ to $z_t^a$ for ease. A prediction $\tilde{z}_t^a$ and its ground-truth $z_t^a$ from the same tuple constitute a positive example, leading to the following loss function definition:

$$L_{bc} = -\log \frac{\exp(\text{sim}(\tilde{z}_t^{a,i}, z_t^{a,i})/\tau)}{\sum_{\substack{j=1 \\ j \neq i}}^{N} \exp(\text{sim}(\tilde{z}_t^{a,i}, z_t^{a,j})/\tau)}, \tag{3}$$

where $\tau$ denotes a temperature parameter that is 0.07 and the mini-batch size $N$ is set to 1024. And $\text{sim}$ means the cosine similarity.

To enable our model with forecasting abilities, upon deriving $z_t^{a,react}$, we employ the pre-trained forward dynamics model to forecast the subsequent state feature $z_{t+1}^{s,react}$, considering the reactor's current state. For simplicity, $z_{t+1}^{s,react}$ is referred to as $z_{t+1}^s$. Here again, we utilize contrastive loss for state supervision, denoted as follows:

$$L_{fd} = -\log \frac{\exp(\text{sim}(\tilde{z}_{t+1}^{s,i}, z_{t+1}^{s,i})/\tau)}{\sum_{\substack{j=1 \\ j \neq i}}^{N} \exp(\text{sim}(\tilde{z}_{t+1}^{s,i}, z_t^{s,j} + 1)/\tau)}, \tag{4}$$

where $\tau$ denotes a temperature parameter that is 0.07 and the mini-batch size $N$ is set to 1024.

So, the final loss function can be defined as:

$$L_{total} = L_{bc} + L_{fd} + L_{reg}. \tag{5}$$

where $L_{reg}$ is expressed as $|a_t^{react}|^2$ to ensure the smoothness of the interaction.

## 4.5 Iterative Generalist-Specialist Learning

The performance of the model plateaus as it forgets older demonstrations when learning new ones. Directly training a unified policy on all action data is extremely challenging, so we employ an Iterative Generalist-Specialist Learning Strategy followed by UniDexGrasp++[51].

Generalist-specialist learning divides the entire task space into smaller subspaces, assigning each subspace to a specialist for focused mastery. This segmentation simplifies learning due to reduced task variations, enabling specialists to excel within their respective domains. Ultimately, the knowledge from all specialists is distilled

into a single generalist. This process, when repeated, is termed Iterative Generalist-Specialist Learning. We employ the state encoder to translate the dataset into feature space, subsequently clustering it into 10 subsets for training our policy via the Iterative Generalist-Specialist Learning strategy.

## 5 EXPERIMENT

### 5.1 Dataset

**InterHuman Datasets.[25]** Following the official guidelines, we've designated 5200 sequences for training and 1177 sequences for testing. We assume that the first human is the actor and the other one is the reactor. We downsample the motion data at 30 fps for training and testing. After the demonstration generation process, we obtained a total of 24,440 training data for training and 5,061 for testing.

**Chi3D Datasets.[13]** Chi3D features a total of 373 available data provided by officials, with 300 allocated for training and 73 for testing. We define the human estimated from images as actors, while the other one captured by motion capture devices is considered as reactors. After the demonstration generation process, we obtained a total of 1,680 training data for training and 313 for testing.

### 5.2 Baselines

For all baseline methods, we utilized the original authors' code, making necessary adjustments to adapt it to our task.

**Progressively Generating Better Initial Guesses**[30] employs Spatial Dense Graph Convolutional Networks and Temporal Dense Graph Networks for enhanced performance.

**Spatio-temporal Transformer**[2] leverages a transformer-based architecture, utilizing attention mechanisms to identify temporal and spatial correlations in human motion prediction.

**InterFormer**[9] features a Transformer network that integrates both temporal and spatial attention to capture the dependencies of interactions across time and space effectively.

**InterGen-revised**[25] is an advanced diffusion-based framework capable of generating multi-human interactions from textual descriptions. We revised the framework by swapping the CLIP branch for a spatio-temporal transformer to encode the actor's motion, focusing on generating multi-human motions. Despite having output and supervision signals for both the actor and reactor, we only utilize the output of the reactor's motions.

**InsActor-revised**[44] is a language-conditioned, physics-based method for generating motion that initiates with a kinematic-based diffusion model for motion creation, subsequently transitioning state to action space. Leveraging the kinematic-based outcomes from InterGen-revised, InsActor calculates the action, standing out as the most relevant baseline by combining cutting-edge kinematics-based diffusion modeling with physics-based tracking.

### 5.3 Metrics

We adopt metrics from previous works on kinematic-based human motion and physically plausible motion generation.

**Fréchet Inception Distance (FVD).** FVD computes the distance between the ground truth and the generated data distribution. We use a pre-trained keypoints-based motion encoder from MotionGPT to extract features from both the generated animations and ground truth motion sequences. And we generate 1000 samples 10 times with different random seeds.

**Diversity Score (Div).** Diversity Score is the average deep feature distance between all the samples. We also generated 1000 samples 10 times here.

**Ground Distance (GD).** We compute the distance between the average floating height (above ground) and the average vertical ground penetration depth when the joint velocity is lower than a threshold in 0.3s (for 10 frames). This is determined by the lowest SMPL-X vertex.

**Interpenetration.** We report the interpenetration volume (IV) of vertices that penetrate the actor mesh and the maximum inter-penetration depth (ID). This metric is computed only when the minimum distance between the actor and the reactor is smaller than 0.2cm. Note that as a consequence of the approximated collision geometry as rigid bodies in the physics simulation, our method can still exhibit small amounts of interpenetration after converting the simulation results to the SMPL-X parameter space.

### 5.4 Evaluation and Discussion

**Compared with existing methods.** We provide qualitative results in 3. Please see our supplementary video for more examples. We also provide quantitative results on InterHuman and Chi3D. The InsActor-revised is a physics-based method, but it relies on the generation quality of kinematic-based methods. All other methods are pure kinematic-based, among which InterGen-revised is the state-of-the-art method. It can be seen that our results are significantly better than existing methods on both FVD and Div metrics. Since both our method and InsActor-revised are physics-based methods, the GD, IV, and ID metrics are all zero in the simulation environment, which also demonstrates the natural advantages of physics-based methods over kinematic-based methods. Because kinematic-based methods lack physical priors, they perform poorly on the GD, IV, and ID metrics. Although InsActor has natural advantages in GD, IV, and ID, it is constrained by the capabilities of kinematic-based methods and needs to resist the noise of kinematic trajectories in subsequent execution phases, such as sliding, floating feet, and penetration, which is not conducive to the learning of policies. Additionally, we found that the noise introduced by kinematic methods significantly affects InsActor's decision-making, making it easy for the reactor to fall during the interaction process. Our method does not rely on the generation quality of kinematic methods but directly learns the stable mapping between states and actions, which can significantly improve performance and generate high-quality reactions.

**Ablations.** We performed ablation studies to assess the impact of the Forward Dynamics Model (FDM), Iterative Generalist-Specialist Learning Strategy (IGSL), and Contrastive Loss (CL). The results show a marked decline in model performance when either the FDM guidance or the IGSL is omitted. Similarly, replacing the CL with L2 loss leads to a substantial performance drop, highlighting the constraints of deterministic loss functions in capturing interactions.

**Robustness with noisy motion capture data.** To demonstrate how InsActor is significantly impacted by noise from kinematic-based methods, we introduced Gaussian noise with variances of 0.01 and 0.05 to the poses in the dataset and observed the performance

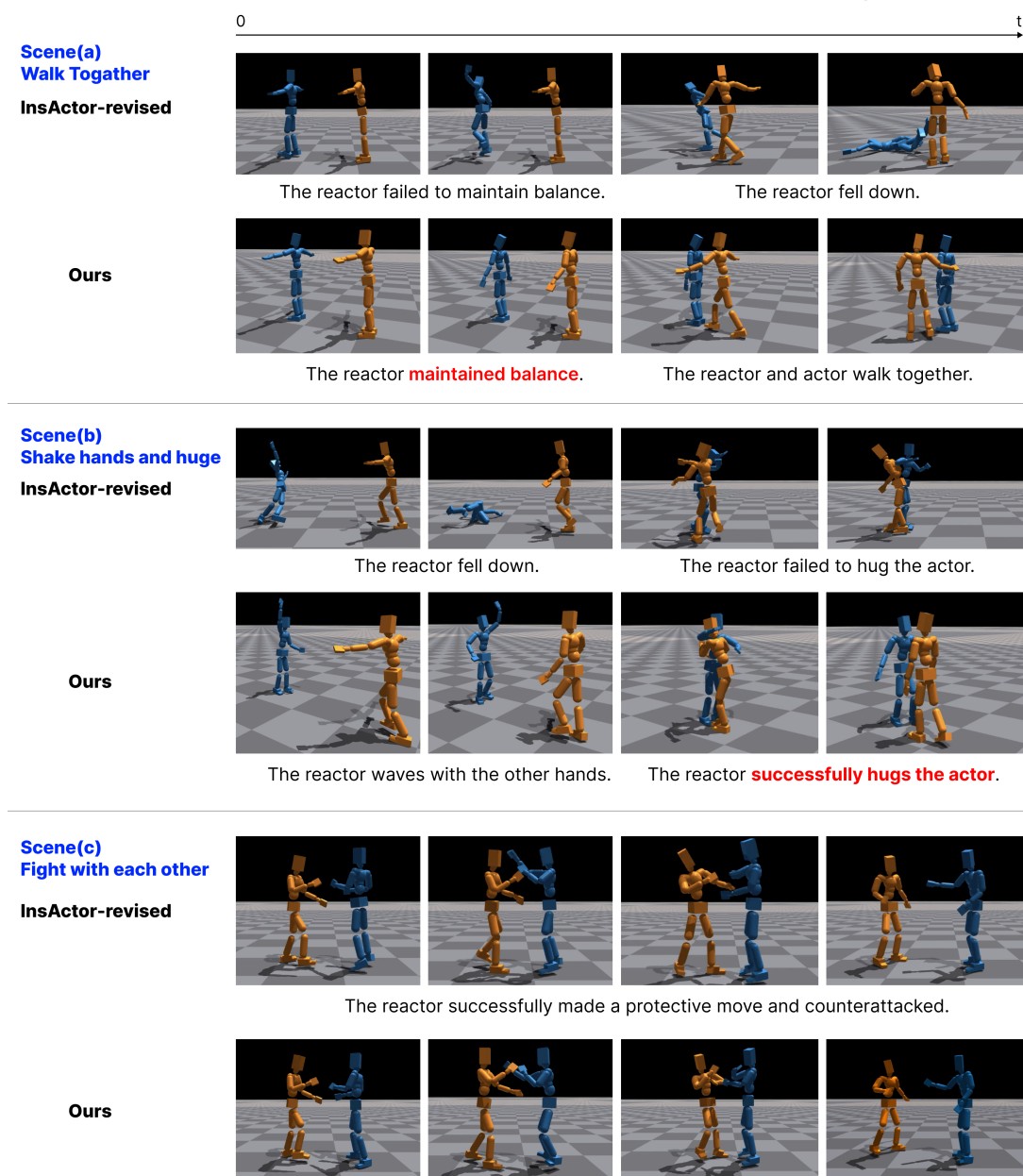

**Figure 3: Qualitative results on InterHuman. It can be observed that our method significantly outperforms the InsActor in terms of stability and realism. While it tends to fall over, our approach can generate stable interactions.**

degradation. Our method remains stable with noise-0.01 and noise-0.05. However, the FVD of InsActor escalates by 4.7 under noise-0.01 and by 18.7 under noise-0.05, underscoring our method's enhanced robustness against motion capture data noise.

**Performance Gap with Small Training Set.** Contrary to kinematic-based approaches that directly predict human poses, our method establishes a stable mapping between interaction states and reactor actions, diminishing the reliance on extensive training data by not needing to learn the intricate interplay of human joint positions and their complex dynamics. Our method showcases a notable superiority by utilizing just 20% training dataset, maintaining exceptional performance against existing methods.

| Method | FVD($\downarrow$) | Div($\rightarrow$) | GD[mm]($\downarrow$) | IV [$cm^3$]($\downarrow$) | ID[$mm$]($\downarrow$) |
|---|---|---|---|---|---|
| | | | InterHuman | | |
| Real | 0.17 | 15.7 | 3.8 | 0.9 | 3.6 |
| PGBIG | 90.2 | 9.7 | 12.3 | 2.1 | 7.2 |
| SS-Transformer | 80.7 | 11.2 | 10.2 | 2.0 | 7.2 |
| InterFormer | 59.4 | 11.8 | 8.6 | 1.7 | 5.4 |
| InterGen-revised | 28.2 | 17.1 | 5.4 | 1.3 | 4.9 |
| InsActor-revised | 30.2 | 13.5 | **0.0/1.1** | **0.0/0.23** | **0.0/1.4** |
| Ours | **14.1** | **15.0** | **0.0/1.1** | **0.0/0.23** | **0.0/1.4** |
| | | | Chi3D | | |
| Real | 0.09 | 12.3 | 5.2 | 1.3 | 4.7 |
| PGBIG | 66.8 | 7.2 | 14.2 | 2.9 | 8.2 |
| SS-Transformer | 77.2 | 9.1 | 11.7 | 2.9 | 6.9 |
| InterFormer | 32.1 | 9.7 | 10.2 | 2.1 | 5.2 |
| InterGen-revised | 22.8 | 14.8 | 6.1 | 1.6 | 5.0 |
| InsActor-revised | 27.1 | 11.1 | **0.0/1.1** | **0.0/0.23** | **0.0/1.4** |
| Ours | **11.4** | **11.6** | **0.0/1.1** | **0.0/0.23** | **0.0/1.4** |

Table 1: Evaluation on InterHuman and Chi3D datasets. Our method significantly outperforms existing methods. For InsActor and our method, the GD, IV, and ID in simulation is 0, while it is 1.1, 0.23, and 1.4 in the SMPL-X space as a consequence of the rigid body approximation of the humanoid in the physics simulation.

| Method | FVD($\downarrow$) | Div($\rightarrow$) |
|---|---|---|
| Real | 0.17 | 15.7 |
| w/o FDM | 28.1 | 17.5 |
| w/o IGSL | 23.4 | 13.2 |
| w/o CL | 19.8 | 14.2 |
| Ours | 14.1 | 15.0 |

Table 2: Ablation. We conducted comprehensive ablation experiments to verify the effectiveness of each component.

| Method | FVD($\downarrow$) | Div($\rightarrow$) |
|---|---|---|
| Real | 0.17 | 15.7 |
| InsActor-0.01 | 34.2$\rightarrow$38.9 | 13.5$\rightarrow$12.5 |
| InsActor-0.05 | 34.2$\rightarrow$52.9 | 13.5$\rightarrow$10.6 |
| Ours-0.01 | 14.1$\rightarrow$14.0 | 15.0$\rightarrow$15.2 |
| Ours-0.05 | 14.1$\rightarrow$17.2 | 15.0$\rightarrow$16.8 |

Table 3: Robustness with noisy motion capture data. We add Gaussian noise to the dataset and then report the performance drops.

**Long-Range Forecasting for Forward Dynamics Models.** To capitalize on the benefits of Forward Dynamics Models, we examined their impact through Long-Range vs. Single-Step Forecasting experiments. Extending the forecast to 50 steps led to a notable decrease in performance, likely due to accumulating errors in Forward Dynamics Models. We also conducted experiments on both Long-Range Forecasting and Single-Step Forecasting simultaneously, and the results show that its performance is not much different from

| Method | FVD($\downarrow$) | Div($\rightarrow$) |
|---|---|---|
| Real | 0.09 | 13.8 |
| InterGen-revised | 32.9 | 14.6 |
| InsActor-revised | 54.7 | 9.7 |
| Ours | 22.6 | 12.6 |

Table 4: Performance Gap with Small Training Set. Our method has a significant advantage and can still achieve advanced performance compared with InterGen-revised and InsActor-revised.

| Method | FVD($\downarrow$) | Div($\rightarrow$) |
|---|---|---|
| Real | 0.17 | 15.7 |
| Step-50 | 24.3 | 16.2 |
| Step-1&50 | 14.9 | 15.4 |
| Step-1(Ours) | 14.1 | 15.0 |

Table 5: Forecasting steps for Forward Dynamics Models. We conducted experiments to investigate the performance differences between Long-Range Forecasting and Single-Step Forecasting

using Single-Step Forecasting alone. Therefore, considering the computational cost, we do not introduce Long-Range Forecasting in the 4D Imitation Learning phase.

**Real-Time Inference.** Our method, instead of using heavy diffusion models, attains real-time inference at 30 fps on a single 3090 GPU. In contrast, InterGen-revised is limited to 0.3 fps, and InsActor reaches only 0.9 fps.

**The Importance of Latent Dynamics Model.** We also experimentally verified the advantages of the Latent Dynamics Model. Our method employs a State/Action VAE to transform the raw data into a feature space, followed by predictions using a forward dynamics model. Directly predicting dynamics on raw human key points and supervising with MSE loss, we found that this significantly increases optimization difficulty and reduces training robustness. Results indicate that on the InterHuman dataset, the FVD can only reach 42.8, demonstrating that the State/Action VAE and Latent Dynamics Model are crucial in our design.

**Limitation.** While our method successfully generates realistic reactions, it comes with limitations. We haven't tested its applicability to scenarios with three or more participants, like basketball games. Currently, it doesn't account for intricate hand movements, such as those in rock-paper-scissors.

## 6 CONCLUSION

This paper presents a Forward Dynamics Guided 4D Imitation method that leverages a forward dynamics model to guide 4D imitation learning. Our approach produces reactions that are physically accurate and human-like reactions. We validated our approach with experiments on the InterHuman and Chi3D datasets, further underscored by extensive ablation studies.

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
