# OpenReview forum: "PhysReaction: Physically Plausible Real-Time Humanoid Reaction Synthesis via Forward Dynamics Guided 4D Imitation"
_acmmm.org/ACMMM/2024/Conference — MM2024 Oral_

### Official Review · Reviewer_mmRR · 2024-05-23

**Rating:** 5
**Confidence:** 2

**Summary:**

This article focuses on the field of human-humanoid interaction and proposes a forward dynamics-guided imitation method to enable the robot to produce reasonable and physically plausible reaction. In order to be deployed on robots, this method is real-time. The effectiveness of the method is verified on two datasets, and there are rich ablation experiments and analysis experiments.

**Strengths:**

1. Real-time robot response actions are achieved, and the method introduces forward dynamics, making the overall action more consistent with physical constraints and more physically plausible.
2. A large number of comparative experiments and ablation experiments were designed to illustrate and verify the effectiveness of the method.

**Limitations:**

All experiments are conducted on a single person. The authors also mention this in the supplementary material. It is indeed difficult to realize the action reaction generation of multiple people. All experiments are deployed in the simulation environment.

**Suitability:**

3

---

### Official Review · Reviewer_quUt · 2024-05-23

**Rating:** 3
**Confidence:** 2

**Summary:**

This paper introduces a Forward Dynamics Guided 4D Imitation method to produce human-like reactions that are both physically plausible and generated in real-time. The proposed method significantly improves the speed and quality of reactions, as demonstrated by experiments on the InterHuman and Chi3D datasets.

**Strengths:**

The method has been evaluated on the InterHuman and Chi3D datasets, showing consistent and significant performance improvements over existing methods. The paper also includes detailed ablation studies and analytical experiments to validate the effectiveness of each component of the proposed approach.

The structure is clear for implementation.

**Limitations:**

There are many typos in the paper, for example in figure 3, is the photo correct for scene c and also It should be hug instead of "huge".

The method is lack of novelty, the structure has widely been used for imitation learning, it's just been applied to your designed tasks.

The method is supposed to be a physics-based method, I did not see that from the paper.

**Suitability:**

3

---

### Official Review · Reviewer_6s6n · 2024-05-24

**Rating:** 4
**Confidence:** 3

**Summary:**

The paper introduces the Forward Dynamics Guided 4D Imitation method, a novel approach that utilizes a neural model to simulate human forward dynamics in real-time at 30 fps, significantly improving the speed for inference. The main goal is to generate physically plausible and human-like reactions for humanoid robots to enhance their interaction with humans. The existing methods, including kinematics-based and physics-based approaches, face limitations in generating realistic motions and real-time inference. The proposed method aims to address these challenges by learning a policy that can generate physically plausible reactions in real-time, bypassing the need for reference states and kinematics-based diffusion models. The key contributions include introducing a new task focused on physics-based online synthesis of humanoid reactive motions, presenting the Forward Dynamics Guided 4D Imitation approach for realistic reactions, and demonstrating superior performance compared to existing methods through experiments on InterHuman and Chi3D datasets.

**Strengths:**

- Introduction of the Forward Dynamics Guided 4D Imitation method for generating physically plausible humanoid reactions.
- Demonstrated superior performance compared to existing methods through experiments on InterHuman and Chi3D datasets.
- Enhanced robustness against motion capture data noise compared to the InsActor baseline method.
- Notable superiority in performance with just 20% of the training dataset, showcasing a stable mapping between interaction states and reactor actions.

**Limitations:**

- Lack of user study for qualitative comparison. Quantitative metrics for human motion generation may not always align with human visual preferences, therefore, a user study is important to demonstrate the qualitative superiority.
- Lack of comparison with more state-of-the-art physics-aware motion generation methods. Only one physics-aware motion generation method is compared in the paper. Here are some recent physics-aware motion generation works:
[1] Learning Physically Simulated Tennis Skills from Broadcast Videos, SIGGRAPH 2023
[2] PhysDiff: Physics-Guided Human Motion Diffusion Model, ICCV 2023 (Oral)
[3] Motion In-betweening for Physically Simulated Characters, SIGGRAPH Asia 2022
- Limited discussion on the generalizability of the proposed method to different types of interactions or datasets, potentially restricting its applicability in broader contexts.
- Other comment. In L634, Fréchet Inception Distance (FVD). The abbreviation FVD usually refers to Frechet Video Distance. Since your feature distribution is based on motion data, you can simply use FMD or FID as utilized in motion generation tasks such as MDM and DiffSHEG.

**Suitability:**

2

---

### Meta-Review · Area_Chair_7Wf5 · 2024-07-08

**Recommendation:** Accept (Oral)
**Confidence:** 5

**Metareview:**

This article has received 3 reviews. The 3 reviewers are confident. The three reviews are positive and highlight the quality of the work... . On this basis, I propose to accept this article